# Depth in convolutional neural networks solves scene segmentation

**Noor Seijdel**[1,2]*, **Nikos Tsakmakidis**[3], **Edward H. F. de Haan**[1,2], **Sander M. Bohte**[3], **H. Steven Scholte**[1,2]

**1** Department of Psychology, University of Amsterdam, Amsterdam, The Netherlands, **2** Amsterdam Brain & Cognition (ABC) Center, University of Amsterdam, Amsterdam, The Netherlands, **3** Machine Learning Group, Centrum Wiskunde & Informatica, Amsterdam, the Netherlands

* noor.seijdel@gmail.com

**Data Availability Statement:** All data files (human behavior) will be made available on the Open Science Framework database (accession number gb89u).

**Funding:** This work was supported by an Advanced Investigator Grant by the European Research

## Abstract

Feed-forward deep convolutional neural networks (DCNNs) are, under specific conditions, matching and even surpassing human performance in object recognition in natural scenes. This performance suggests that the analysis of a loose collection of image features could support the recognition of natural object categories, without dedicated systems to solve specific visual subtasks. Research in humans however suggests that while feedforward activity may suffice for sparse scenes with isolated objects, additional visual operations ('routines') that aid the recognition process (e.g. segmentation or grouping) are needed for more complex scenes. Linking human visual processing to performance of DCNNs with increasing depth, we here explored if, how, and when object information is differentiated from the backgrounds they appear on. To this end, we controlled the information in both objects and backgrounds, as well as the relationship between them by adding noise, manipulating background congruence and systematically occluding parts of the image. Results indicate that with an increase in network depth, there is an increase in the distinction between object- and background information. For more shallow networks, results indicated a benefit of training on segmented objects. Overall, these results indicate that, de facto, scene segmentation can be performed by a network of sufficient depth. We conclude that the human brain could perform scene segmentation in the context of object identification without an explicit mechanism, by selecting or "binding" features that belong to the object and ignoring other features, in a manner similar to a very deep convolutional neural network.

## Author summary

To what extent do Deep Convolutional Neural Networks exhibit sensitivity to scene properties (e.g. object context) during object recognition, and how is this related to network depth? Linking human visual processing to performance of feed-forward DCNNs with increasing depth, our study explored if and how object information is differentiated from the backgrounds they appear on. We show that with an increase in network depth, there is a stronger selection of parts of the image that belong to the target object, compared to the rest of the image. In other words, network depth facilitates scene segmentation. Given

Council (ERC grant FAB4V #339374) to EdH. The funders had no role in study design, data collection and analysis, decision to publish, or preparation of the manuscript.

**Competing interests:** The authors have declared that no competing interests exist.

that the operations of a very deep network can be performed by a recurrent network, we speculate that the human brain could perform scene segmentation, in the context of object identification, without an explicit mechanism using recurrent processing.

## Introduction

Visual object recognition is so swift and efficient that it has been suggested that a fast feed-forward sweep of perceptual activity is sufficient to perform the task [1–3]. Disruption of visual processing beyond feed-forward stages (e.g. >150 ms after stimulus onset, or after activation of higher order areas) can however lead to decreased object recognition performance [4,5], and a multitude of recent findings suggest that while feed-forward activity may suffice to recognize isolated objects that are easy to discern, the brain employs increasing feedback or recurrent processing for object recognition under more 'challenging' natural conditions [6–10]. When performing a visual object recognition task, the visual input (stimulus) elicits a feed-forward drive that rapidly extracts basic image features through feedforward connections [11]. For sparse scenes with isolated objects, this set of features appears to be enough for successful recognition. For more complex scenes, however, the jumble of visual information ('clutter') may be so great that object recognition cannot rely on having access to a conclusive set of features. For those images, extra visual operations ('visual routines'), such as scene segmentation and perceptual grouping, requiring several iterations of modulations and refinement of the feedforward activity in the same and higher visual areas, might be necessary [11–14].

While this view emphasises that object recognition relies on the integration of features that belong to the object, many studies have shown that features from the background can also influence the recognition process [15–22]. For example, objects appearing in a familiar context are detected more accurately and quickly than objects in an unfamiliar environment, and many computational models of object recognition (in both human and computer vision), use features both from within the object and from the background [23–25]. This shows that when subjects recognise an object, figure-ground segmentation has not always occurred completely.

One way to understand how the human visual system processes information involves building computational models that account for human-level performance under different conditions. Here we investigate Deep Convolutional Neural Networks (DCNNs). DCNNs are being studied often because they show remarkable performance on both object and scene recognition, rivaling human performance. Recent evidence shows that the depth of DCNNs is of crucial importance for this recognition performance [26]. In addition to better performance, deeper networks have also been shown to be more human-like (making errors similar to human subjects; [27]). More layers seem especially important when scenes are more difficult or challenging, e.g. because of occlusion, variation, or blurring, where elaborate processing is required [8,10]. The very deep residual networks used in current object recognition tasks are nearly equivalent to a recurrent neural network unfolding over time, when the weights between their hidden layers are clamped [28]. This has led to the hypothesis that the additional layers function in a way that is similar to recurrent processing in the human visual system, and that these additional layers are solving the challenges that are resolved by recurrent computations in the brain.

In the current study, we explore how the number of layers (depth) in a DCNN relates to human vision and how depth influences to what degree object segmentation occurs. While we certainly do not aim to claim that DCNNs are identical to the human brain, we argue that they can be studied in a way similar to the way in which we use animal models (DNimals; [29]).

First, we focused on the question to what extent DCNNs exhibit the same sensitivity to scene properties (object context) as human participants. To this end, we presented seven Residual Networks (ResNets; [30]) with an increasing number of layers and 40 human participants with images of objects that were either presented on a uniform background (segmented), or on top of congruent or incongruent scenes and evaluated their performance. Additionally, for the DCNNs, we controlled the amount of information in the objects and backgrounds, as well as the relationship between them by adding noise or systematically occluding parts of the image. Next, we investigated the role of segmentation on learning ('training'), by training the DCNNs on either segmented or unsegmented objects.

A convergence of results indicated a lower degree of segregation between object- and background features in more shallow networks, compared to deeper networks. This was confirmed by the observation that more shallow networks benefit more from training on pre-segmented objects than deeper networks. Overall, deeper networks seem to perform implicit 'segmentation' of the objects from their background, by improved selection of relevant features.

## Results

### Experiment 1: Scene segmentation + background consistency effect

**Human performance.**   In experiment 1, participants viewed images of real-world objects placed onto white (segmented), congruent and incongruent backgrounds (Fig 1). Images were presented in randomized sequence, for a duration of 32 ms, followed by a mask, presented for 300 ms. After the mask, participants indicated which target object was presented, by clicking on one of 27 options on screen using the mouse (see Materials and methods).

Accuracy (percentage correct) was computed for each participant. A non-parametric Friedman test differentiated accuracy across the three conditions (segmented, congruent, incongruent), Friedman's Q(2) = 74.053, $p < .001$. Post-hoc analyses with Wilcoxon signed-rank tests indicated that participants made fewer errors for segmented objects, than the congruent, W = 741, $p < .001$, and incongruent condition, W = 741, $p < .001$. Additionally, participants made fewer errors for congruent than incongruent trials, W = 729, $p < .001$. Overall, results indicate that when a scene is glanced briefly (32 ms, followed by a mask), the objects are not completely segregated from their background and semantic consistency information influences object perception.

**Model performance.**   For human participants, results indicated that (at a first glance) features from the background influenced object perception. Do DCNNs show a similar pattern and how is this influenced by network depth? To investigate the effect of network depth on scene segmentation, tests were conducted on seven deep residual networks (ResNets; [29]) with increasing number of layers: 6 (technically, this network is a "Net6", because we removed the residual connection), 10, 18, 34, 50, 101 and 152.This approach allowed us to investigate the effect of network depth (adding layers) while keeping other model properties as similar as possible.

We presented 38 different subsets of 243 stimuli to the DCNNs, each subset consisting of the same number of images per category and condition that human observers were exposed to (81 per condition, 3 per category). Following the procedure for comparing human performance, a non-parametric Friedman test differentiated accuracy across the three conditions (segmented, congruent, incongruent) for all networks. Using Post-hoc Wilcoxon signed-rank tests with Benjamini/Hochberg FDR correction, differences between the conditions were evaluated for all networks (Fig 2; significant differences indicated with a solid line).

Results indicated both a substantial overlap and difference in performance between human participants and DCNNs (Fig 2). Both were better in recognizing an object on a congruent

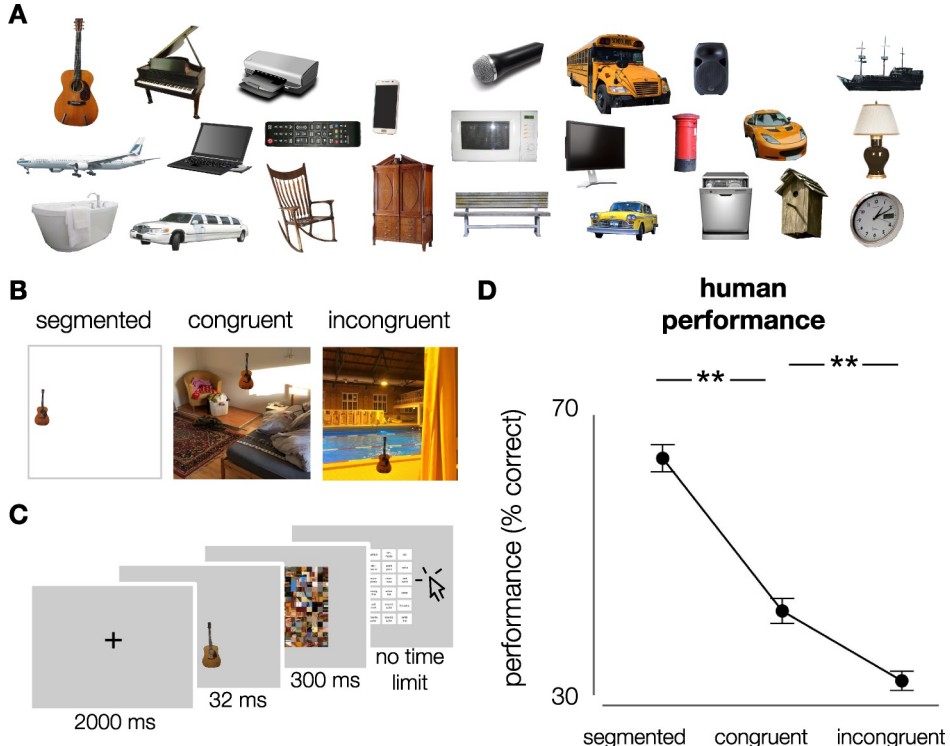

**Fig 1. Stimuli and experimental design. A)** Exemplars of the different object categories (cut-out objects from ImageNet validation set). 27 object categories were used in this experiment (subordinate level, based on ImageNet categories). In total, each category contained 10 exemplars. **B)** Stimuli were generated by placing the objects onto white, congruent and incongruent backgrounds (512*512 pixels, full-color). Backgrounds were sampled from the SUN2012 database [53]. For human participants, objects were downsized and placed in one of nine possible locations (3x3 grid). For DCNNs, objects were bigger and placed centrally. **C)** Participants performed on an object recognition task. At the beginning of each trial, a fixation-cross was presented in the center of the screen for 2000 ms, followed by an image. Images were presented in randomized sequence, for a duration of 32 ms, followed by a mask, presented for 300 ms. After the mask, participants had to indicate which object they saw, by clicking on one of 27 options on screen using the mouse. After 81 (⅓) and 162 (⅔) trials, there was a short break. **D)** Human performance (% correct) on the object recognition task. Participants performed best for segmented objects, followed by congruent and incongruent respectively. Error bars represent bootstrap 95% confidence intervals.

versus an incongruent background. However, whereas human participants performed best in the segmented condition, DCNNs performed equally well (or better) for the congruent condition. Performance for the incongruent condition was lowest. This effect was particularly strong for more shallow networks (ResNet-6, ResNet-10), and got smaller as the networks got deeper. A Mann-Whitney U test on the difference in performance between congruent and incongruent trials indicated a smaller decrease in performance for incongruent trials for ResNet-152 compared to ResNet-6 (Mann–Whitney U = 1420.0, n1 = n2 = 38, $p < .001$, two-tailed) For 'ultra-deep' networks it mattered less if the background was congruent, incongruent or even present, behavior that humans also exhibit when these images are shown unmasked. Remarkably, performance of the most shallow network (ResNet-6) was better for the congruent condition compared to the segmented condition. These results suggest that parts of co-varying backgrounds or surroundings influence the categorization of the objects. In other words, there is 'leakage' of the natural (congruent) background in the features for classification, predominantly for the more shallow networks. For object recognition in a congruent scene this is not necessarily a problem, and can even increase or facilitate performance (as compared to the segmented condition). For objects on an incongruent background, however, this impairs

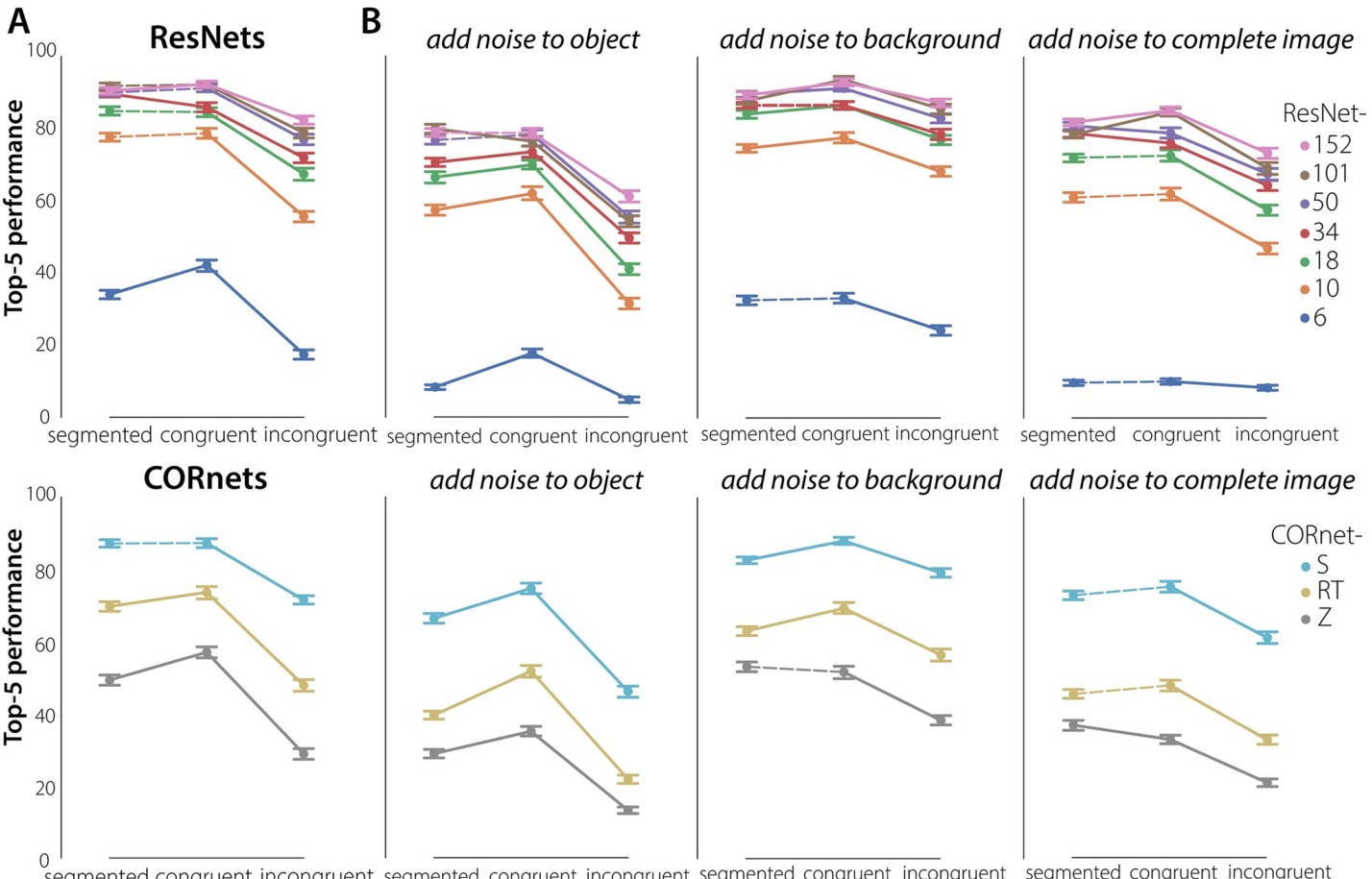

**Fig 2. DCNN performance on the object recognition task. A)** DCNN performance on the object recognition task. 38 different subsets of 243 stimuli were presented, each subset consisting of the same number of images per target category and condition (segmented, congruent, incongruent) that human observers were exposed to (81 per condition, 3 per category). For all models, performance was better for the congruent than for the incongruent condition. For the ResNets, this decrease was most prominent for ResNet-6, and got smaller as the models get deeper. For 'ultra-deep' networks it mattered less if the background was congruent, incongruent or even present. For the CORnets, this decrease was most prominent for the feedforward architecture (CORnet-Z). For CORnet-S (recurrent + skip connection) performance was similar to an 'ultra-deep' network. Using Post-hoc Wilcoxon signed-rank tests with Benjamini/Hochberg FDR correction, differences between the conditions were evaluated for all networks. Significant differences are indicated with a solid line vs. a dashed line (all segmented–incongruent comparisons were significant). Error bars represent bootstrap 95% confidence interval. **B)** DCNN performance on the object recognition task after adding noise to the object, the background, or both.

classification performance. These results suggest that one of the ways in which network depth improves object classification, is by learning how to select the features that belong to the object, and thereby implicitly segregating the object features from the other parts of the scene.

Then, to determine whether the experimental observations above can be approximated by recurrent computations, we additionally tested three different architectures from the CORnet model family[31]; CORnet-Z (feedforward), CORnet-R (recurrent) and CORnet-S (recurrent with skip connections). The shift in performance from CORnet-Z to CORnet-S showed the same pattern as the shift from ResNet-6 to ResNet-18. This overlap suggests that the pattern of results for deeper ResNets can be approximated by recurrent computations. Because the different CORnet models did not only differ with respect to 'recurrence', but also contained other architectural differences (CORnet-Z not only is feedforward, but it is also shallower than COR-net-S), the differences between the networks could stem from the difference in information flow (feedforward vs. recurrent), or from the different amount of parameters in each network. Taking the results from the ResNets and CORnets together, these findings suggest that one of

the ways in which network depth improves object classification, is by learning how to select the features that belong to the object, and thereby implicitly segregating the object features from the other parts of the scene.

To confirm this hypothesis, and to gain more insight into the importance of the features in the object vs. the background, Gaussian noise was added to either the object, the background, or both (Fig 2B). When noise was added to the complete image (object included), performance decreased for all conditions and all networks. When noise was added to the object only, classification performance also decreased for all conditions Crucially, this decrease was modest for the congruent and particularly severe for the incongruent condition. This indicates that for the congruent condition, also in the no-noise manipulation, performance is heavily dependent on the background for classification. The other side of this conclusion, that in the incongruent condition the features in the background interfere with object classification, is confirmed by the observation that this condition improves when noise is added to the background.

To further investigate the degree to which the networks are using features from the object and/or background for classification, we systematically occluded different parts of the input image by sliding a gray patch of either 128*128 (Fig 3), 64*64 or 256*256 pixels (Fig 4) across the image in 32 pixel steps. We evaluated the changes in activation of the correct class after occlusion of the different image parts, before the softmax activation function (compared to activation for the 'original' unoccluded image). We reasoned that, if the activity in the feature map changed after occluding a patch of the image, that those pixels were important for classification. For this analysis, positive values indicate that pixels are helping classification, with higher values indicating a higher importance. This reveals the features to be far from random, uninterpretable patterns. For example, in Fig 3, results clearly show that the network is localizing the object within the scene, as the activity in the feature map drops significantly when the object (china cabinet in this example) is occluded. To evaluate whether deeper networks are better at localizing the objects in the scene, while ignoring irrelevant background information, we quantified the importance of features in the object vs. background by averaging the change in the feature map across pixels belonging to either the object or the background ('importance'). For each image, importance values of the objects and backgrounds were normalized by dividing them by the activation for the original image. Because performance of ResNet-6 for the 'original' unoccluded images was already exceptionally low, the averaged interference was hard to interpret and remained low, due to many near-zero values in the data. Therefore, we took into account only images that were classified correctly (correct class within Top 5 predictions), resulting in an unequal number of images for each network. Mann-Whitney U tests with Benjamini/Hochberg FDR correction indicated a smaller influence (importance) of background pixels on classification for deeper networks. For those models, pixels from the object had a smaller impact as well, for the segmented and congruent condition.

Next, we tested how training was influenced by network depth. If deeper networks indeed implicitly learn to segment object from background, we expect them to show a smaller difference in learning speed, when trained with segmented vs. unsegmented stimuli (as compared to shallow networks).

## Experiment 2: Training on unsegmented/segmented objects

Experiment 1 indicated that, when trained on ImageNet, network performance is influenced by visual information from both the object and the background region. In experiment 2, we investigated the influence of background on classification performance when the networks are trained on visual information from the object region only. To do so, we trained four networks (ResNet-6, ResNet-10, ResNet-18, ResNet-34) on a dataset with objects that were already

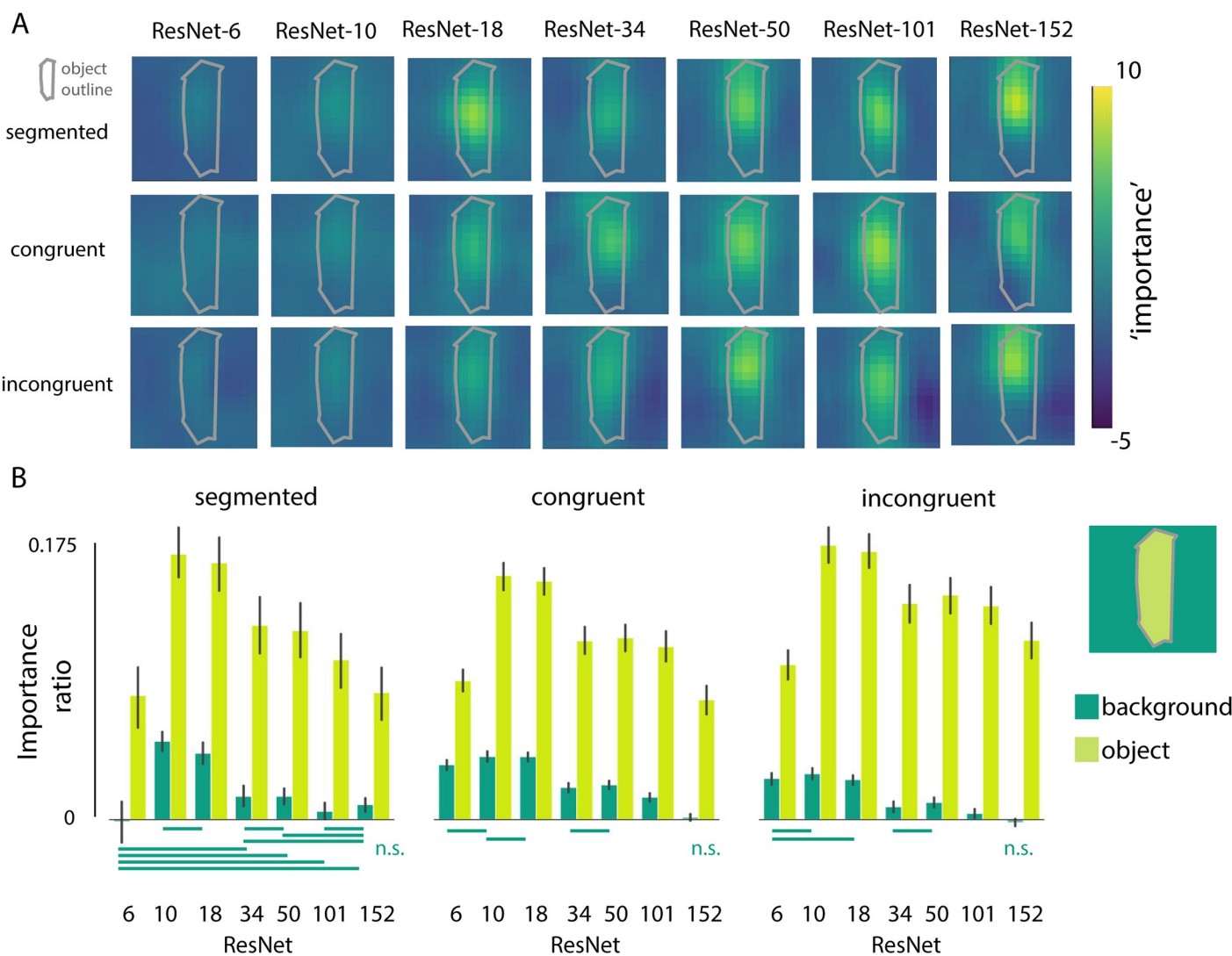

**Fig 3. Systematic occlusion of parts of the image. A)** Examples where we occluded different portions of the scene, and visualized how the classifier output for the correct class changed (before the softmax activation function). Images were occluded by a gray patch of 128x128 pixels, sliding across the image in 32 pixel steps. Importance is defined as the relative change in activation after occluding that part of the image (compared to the activation of the 'original' unoccluded image) and is computed as follows: original activation—activation after occlusion / original activation. This example is for illustrative purposes only; maps vary across exemplars. **B)** The relative change in activation (compared to the original image), after occluding pixels of either the object or the background, for the different conditions (segmented, congruent, incongruent). For each image, importance values of the objects and backgrounds were normalized by dividing them by the activation for the original image, resulting in the importance ratio. Error bars represent the bootstrap 95% confidence interval. Non-significant differences are indicated with a solid line below the graph.

segmented, and on a dataset in which they were unsegmented (i.e. objects embedded in the scene). All images were resized to 128x128 pixels. We used more shallow networks and fewer object classes to reduce computation time. To obtain statistical results, we reinitialized the networks with different seeds and repeated the process for 10 different seeds.

Accuracy of the ResNets was evaluated after each epoch (100 in total) on the validation sets. Results indicated a higher classification accuracy in the early stages of training for the networks trained on segmented objects compared to the networks trained on unsegmented objects (Fig 5). Statistical analyses comparing the average accuracies of the first 10 epochs for networks trained on segmented vs. unsegmented objects indicated significant differences for all models (Mann-Whitney U-statistic: U = 5.0, $p < .001$, for ResNet-6, -10, -18 and -34 respectively).

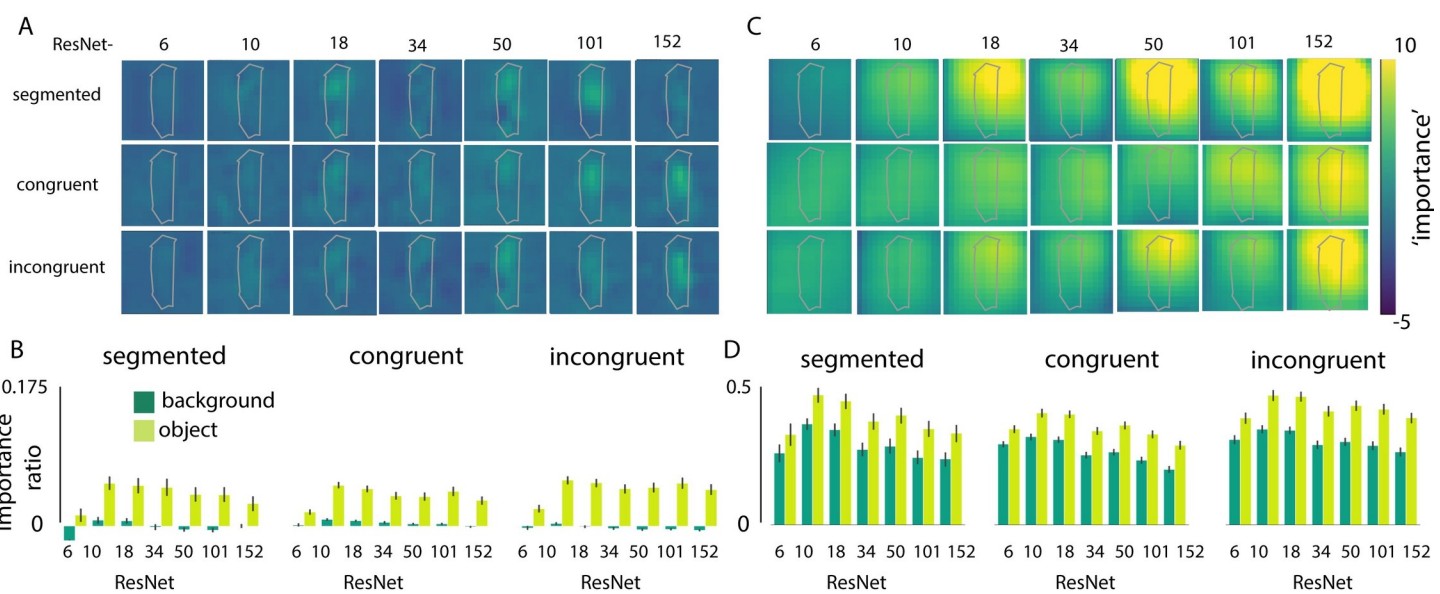

**Fig 4. Analysis repeated with a smaller (64x64) and larger (256x256) patch. A)** visualization of the change in classifier output for the correct class, before the softmax activation function after occlusion by a 64x64 patch, sliding across the image in 32 pixel steps. **B)** The relative change in activation (compared to the original image), after occluding pixels of either the object or the background, for the different conditions (segmented, congruent, incongruent). For each image, importance values of the objects and backgrounds were normalized by dividing them by the activation for the original image, resulting in the importance ratio. Error bars represent the 95% confidence interval. **C/D)** Repeated for a large patch (256x256 pixels).

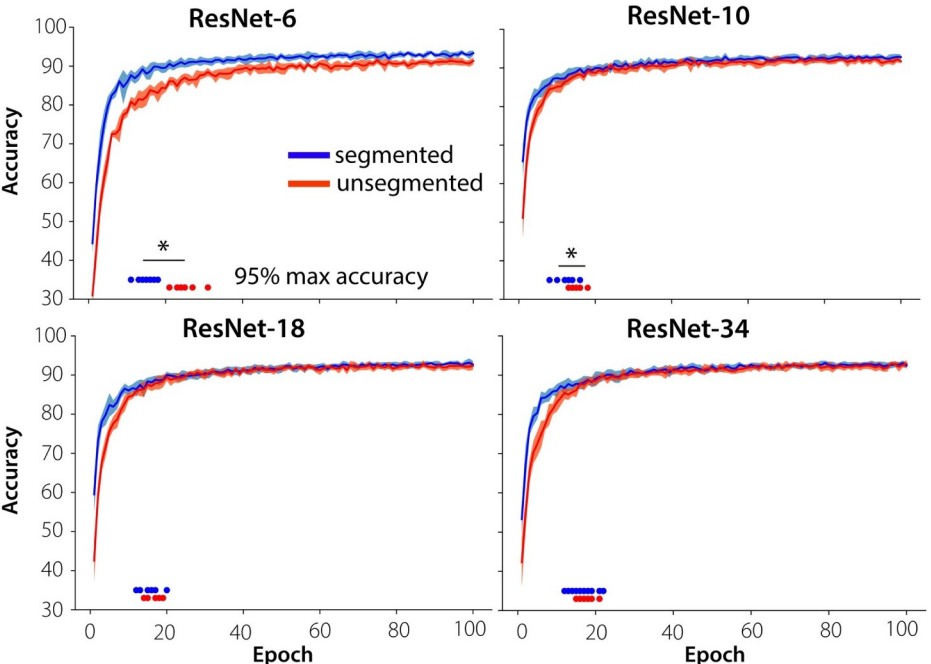

**Fig 5. Accuracy during training on segmented vs. unsegmented stimuli.** Networks trained on segmented objects achieve better classification accuracy in the early stages of training than the networks trained on unsegmented objects for shallow networks (ResNet-6, ResNet-10), and they converge in less epochs. Individual data points indicate the moment of convergence, defined as the first epoch to reach 95% of the maximum accuracy across all epochs.

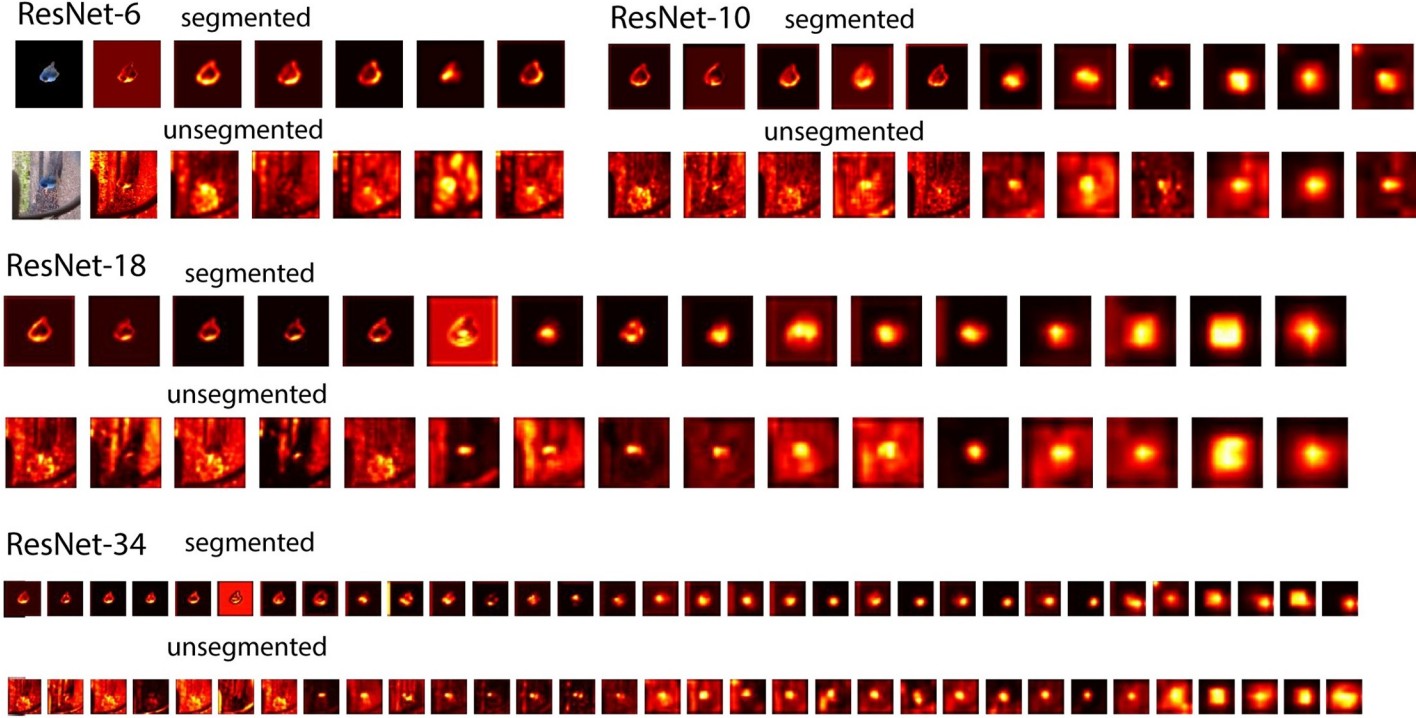

**Fig 6. Visualization of the filter activations of each convolution layer for the different networks.** All the filter activations from the different layers (one 2D-array per filter) for a specific image were extracted. heatmaps were generated by summing the absolute value of those arrays together. The lightest part of these heatmaps contain the most important features for classification. Maps for ResNet-34 were resized for visualization purposes.

In the later stages, accuracy of the two types of models (trained on unsegmented vs. segmented) was similar. Results also indicated a difference between the more shallow networks (ResNet-6), where there is a difference in accuracy between segmented and unsegmented objects for all training epochs, and the deeper networks. For the deeper networks, the difference in accuracy quickly diminishes and finally disappears. Shallow networks trained on segmented stimuli also converged (stabilized) earlier than when they were trained on unsegmented images. Statistical analyses comparing the 'speed of convergence' indicated significant effects of visual training diet (segmented vs. unsegmented) across multiple initialization conditions of the networks, for the more shallow networks (Mann Whitney-U statistic $U = 0$, $p < .001$; $U = 20.0$, $p = .012$ for ResNet6 and ResNet10, respectively). For this analysis, the speed of convergence was defined as the first epoch at which 95% of the maximum accuracy was reached. Deeper networks thus seem to learn to 'segment' the objects from their background during training.

To better understand the inner workings of our models, we visualized the filter activations of each convolution layer for one initialization. Visualizing the filter activations of each convolution layer of the networks provides us with heatmaps that show features of a given image, that a corresponding filter is tuned to. This gives an idea of which parts of the image contained the most important features for classification. To obtain these heatmaps, we extracted all the filter activations from the different layers (one 2D-array per filter) for a specific image. Then, for each layer, we summed the absolute value of those arrays together.

Looking at the heatmaps of networks trained on segmented vs. unsegmented data (Fig 6), we see that the heatmaps of the networks trained on segmented objects contain no background activations. For networks trained on unsegmented objects (full images), however, we see that

the backgrounds are gradually suppressed inside the network. This indicates that the networks learn to attend to important features (i.e. the objects) and almost eliminate completely the influence of the background, when the depth or capacity of the network is sufficient. This suggests that the network learns to segment the objects before classifying.

## Discussion

We investigated the extent to which object and context information is represented and used for object recognition in trained deep convolutional neural networks. Experiment 1 showed both a substantial overlap, and a difference in performance between human participants and DCNNs. Both humans and DCNNs are better in recognizing an object on a congruent versus an incongruent background. However, whereas human participants performed best in the segmented condition, DCNNs performed equally well (or better) for the congruent condition. Performance for the incongruent condition was lowest. This effect was particularly strong for more shallow networks. Further analyses, investigating which parts of the image were most important for recognition, showed that the influence of the background features on the response outcome was relatively strong for shallow networks and almost absent for deeper networks. For shallow networks, the results of experiment 2 indicated a benefit of training on segmented objects (as compared to unsegmented objects). For deeper networks, this benefit was much less prominent. Training on segmented images thus reduced the difference in performance between shallow and deeper networks.

The current results suggest that there is no discrete 'moment' at which segmentation is successful or 'done'. We interpret these findings as indicating that with an increase in network depth there is better selection of the features that belong to the output category (vs. the background), resulting in higher performance during recognition. Thus, more layers are associated with 'more' or better segmentation, by virtue of increasing selectivity for relevant constellations of features. This process is similar, at least in terms of its outcome, to figure-ground segmentation in humans and might be one of the ways in which scene segmentation is performed in the brain using recurrent computations.

### Explicit vs. implicit models of grouping and segmentation

Classic models focussing on grouping and segmentation presume an explicit process in which certain elements of an image are grouped, whilst others are segregated from each other, by a labelling process [32,33]. Several studies have established the involvement of such explicit grouping mechanisms during specific visual tasks. For example, different curve tracing paradigms require grouping of spatially separate contour segments [34], and recent findings by Doerig, Bornet, Rosenholtz, Francis, Clarke & Herzog [35], comparing a wide range of computational models, indicate that an explicit grouping step is crucial to explain different (un)crowding phenomena. Adding explicit segmentation mechanisms to DCNNs is promising to explain human behavior in tasks that require integrating and grouping of global features, or shape-level representations. Our results from behavioral experiments with segmented and unsegmented objects show that when the task is *object recognition* an explicit segmentation step is typically not necessary. We show that with an increase in network depth, there is a stronger influence of the features that belong to the object on recognition performance, showing that 'implicit' segmentation occurs. When this process becomes more efficient (with a deeper network, or recurrent processing) the result is a situation in which, just as in 'explicit' segmentation, the network (or visual system) knows which features belong together, and which ones do not.

Previous studies have already looked into DCNN performance on unsegmented images [36,37], or have even shown a decrease in classification accuracy for unsegmented, compared to segmented objects [27]). In those images, however, objects were placed on a random background, thereby often incongruent (or coincidentally, congruent). In the current study, by manipulating the relevance and usefulness of the background information, we could disentangle whether this decrease was due to a segmentation problem, or the presence of incongruent, misleading information.

## Contextual effects in object recognition

Different accounts of object recognition in scenes propose different loci for contextual effects [38,39]. It has been argued that a bottom-up visual analysis is sufficient to discriminate between basic level object categories, after which context may influence this process in a top-down manner by priming relevant semantic representations, or by constraining the search space of most likely objects (e.g. [40]). Recent studies have also indicated that low-level features of a scene (versus high-level semantic components) can modulate object processing [39] by showing that seemingly meaningless textures with preserved summary statistics contribute to the effective processing of objects in scenes. Comparably, in the current study the DCNNs were agnostic to the meaning of the backgrounds, as they were not trained to recognize, for example, kitchens or bedrooms. The current results show that visual context features may impact object recognition in a bottom-up fashion, even for objects in a spatially incongruent location.

Previous studies have indicated that explicitly augmenting DCNNs with human-derived contextual expectations (likelihood, scale and location of a target object) was able to improve detection performance, potentially indicating a difference in contextual representations in the networks and the humans [41]. In the current study, findings show that only training DCNNs on a large dataset (ImageNet), enables them to learn human-like contextual expectations as well.

## Feed-forward vs. recurrent processing

Instead of being an ultra-deep feed-forward network, the brain likely uses recurrent connections for object recognition in complex natural environments. There are a multitude of findings that have firmly established the involvement of feedback connections during figure-ground segmentation. For example, behavior and neural activity in V1 evoked by figure-ground stimuli are affected by backward masking [42], region-filling processes that are mediated by feedback connections lead to an enhanced neural representation for figure regions compared to backgrounds in early visual areas [43], responses by neurons showing selectivity to border ownership are modulated depending on the location of a 'figure' relative to other edges in their receptive field [44], and the accuracy of scene segmentation seems to depend on recurrent connections to sharpen the local elements within early visual areas [45] (and there are many more). The current results do not speak to those findings, but merely indicate that a very deep feed-forward architecture is capable of obtaining a 'segmented' representation of an object, without recurrent projections.

The interpretation that deeper networks are better at object recognition, because they are capable of limiting their analysis to (mostly) the object–when necessary–is consistent with the idea that deeper networks are solving the challenges that are resolved by recurrent computations in the brain [28]. Previous findings comparing human behavior or the representational geometry of neural responses to DCNNs (e.g. [36,46]) often use images that contain (mostly) frontal views of objects on uniform backgrounds. For segmented objects, on a white or

uniform background, all incoming information is relevant and segmentation is not needed. For those scenes, feed-forward activity in the brain may suffice to recognize the objects [6]. In line with those findings, we also see that even very shallow networks are able to perform well on those scenes. For more complex scenes, on the other hand, the first feed-forward sweep might not be not sufficiently informative, and correctly classifying or recognizing the object might require additional processing. For those scenes, we see a decrease in classification performance, mainly for the more shallow networks. These findings are in line with the global-to-local (or coarse-to-fine) processing framework, in which a coarse visual representation is acquired by the initial feedforward sweep. If this coarse representation is not informative enough to solve the task at hand, additional, more sophisticated visual processes ('routines') can be recruited to refine this representation [6,11,12,47–50]).

## Background congruency

In human natural vision, extraction of gist can lead to a set of expectations regarding the scene's composition, indicating the probability of the presence of a certain object in a scene, but also its most probable locations [20,22]. In the current study, in incongruent scenes, objects did not only violate the overall meaning of the scene category (semantic violation), but were also placed in a position that was not predicted by the local structure of the scene (syntactic violation). On top of that, objects in the human categorization task were placed in a semi-random location across trials to make the task more difficult. This spatial uncertainty, however, has the additional benefit that it makes the task more comparable to the task we ask DCNNs to perform, as DCNNs have no knowledge about the spatial location. A pilot study using stimuli with centered 3D-rendered objects indicated no difference in performance between congruent and incongruent images (see S1 Fig). While this is contrary to published literature [51], there are several factors that might explain this difference. First of all, we used 3D-rendered, computer generated objects, placed on natural scenes (real-world pictures). The difference in visual quality and 'style' between the object and the background might have influenced perception, by making it easier to distinguish them from each other. A second reason might be the size of the objects. Compared to the stimuli used by Davenport and Potter [16] or Munneke et al. [51], our objects were quite large, in order to obtain good network performance.

## Conclusion

With an increase in network depth there is better selection of the features that belong to the output category. This process is similar, at least in terms of its outcome, to figure-ground segmentation in humans and might be one of the ways in which scene segmentation is performed in the brain.

## Materials and methods

### Experiment 1 (scene segmentation and background consistency)

**Ethics statement.**   All participants provided written informed consent and were rewarded with research credits or received a monetary compensation. The experiment was approved by the ethics committee of the University of Amsterdam.

**Participants.**   40 participants (9 males) aged between 18 and 30 years ($M$ = 22.03, $SD$ = 3.02) with normal or corrected-to-normal vision, took part in the experiment. Data from the first two participants were excluded from further data analyses due to technical problems.

**Networks.**   We used Residual Networks (ResNets; [29]) as a method to systematically manipulate network depth because this type of network consists of a limited number of fixed

components that can be up-scaled without altering the architecture in another way. To evaluate whether the performance of ultra-deep ResNets can be approximated by recurrent computations, we al;so tested three different architectures from the CORnet model family (Kubilius et al., 2018); CORnet-Z (feedforward), CORnet-R (recurrent) and CORnet-S (recurrent with skip connections). Our implementation uses the PyTorch deep learning framework [52] and the torchvision package. ResNet-6 was trained on ImageNet [26] with 1 GPU. The other ResNets were downloaded (pretrained).

**Stimuli.**   Images of 27 different object categories were generated by placing cut-out objects from the ImageNet validation set onto white (segmented), congruent and incongruent backgrounds. The categories were defined at a (sub)ordinate level, based on ImageNet categories: acoustic guitar, airliner, bathtub, birdhouse, cab, canoe, cellular telephone, china cabinet, dishwasher, grand piano, laptop, limousine, loudspeaker, mailbox, microphone, microwave, park bench, pirate ship, printer, remote, rocking chair, schoolbus, screen, speedboat, sports car, table lamp, wall clock (Fig 1A). There were ten exemplars for every object category. Backgrounds were sampled from a large database of images obtained from the SUN2012 database [53] (512*512 pixels, full-color). For each category, three typical backgrounds were selected using the five most common places where this object was found within the database (sorted by number of instances inside each scene type). Three atypical backgrounds were manually chosen (Fig 1B). In total, the stimulus set contained 810 images with a congruent background, 810 with an incongruent background and 270 images with segmented objects.

To familiarize human participants with the categories, one of the ten exemplars for each category was randomly selected and used in a practice-run. Using the remaining nine exemplars—three for each condition (segmented, congruent, incongruent) - 243 images were generated for the actual experiment. Each exemplar was only presented once for each participant. To ensure participants processed the complete image, exemplars were downsized and placed in one of 9 possible locations (3x3 grid). Importantly, to rule out any effect of 'exemplar-complexity' (e.g. one guitar being easier to recognize than another) or an interaction between the object, location and the background, all possible exemplar-background-location combinations were balanced over participants.

For DCNNs, to make the comparison with human participants more valid and to estimate the reliability of the effects in our experiment, we showed different subsets of 243 stimuli to the DCNNs, each subset consisting of the same number of images per category and condition that human observers were exposed to (81 per condition, 3 per category).

**Experimental procedure.**   Participants performed on an object recognition task (Fig 1C). At the beginning of each trial, a fixation-cross was presented for 2000 ms, followed by an image. Images were presented in randomized sequence, for a duration of 32 ms, followed by a mask. The masks consisted of scrambled patches of the images and was presented for 300 ms. After the mask, participants had to indicate which object they had seen, by clicking on one of 27 options on screen using the mouse. After 81 (⅓) and 162 (⅔) trials, there was a short break. Using this paradigm, our human object recognition task was closely analogous to the large-scale ImageNet 1000-way object categorization for which the DCNNs were optimized and thus expected to perform well.

**Statistical analysis: Human performance.**   Accuracy (percentage correct) was computed for each participant. Differences in accuracy between the three conditions (segmented, congruent, incongruent) were statistically evaluated using a non-parametric Friedman test. A significant main effect was followed up by Wilcoxon signed-rank tests. Data were analyzed in Python.

**Statistical analysis: DCNNs.**   For each of the images, the DCNNs (ResNet-6, ResNet-10, ResNet-18, ResNet-34, Resnet-50, ResNet-101, ResNet-152) assigned a probability value to

each of the 1000 object categories it had been trained to classify. For each condition (segmented, congruent, incongruent) the Top-5 Error (%) was computed (classification is correct if the object is among the objects categories that received the five highest probability assignments). Then, to gain more insight in the importance of the features in the object vs the background for classification, we added Gaussian noise to either the object, background, or to both (the complete image) and evaluated performance.

## Experiment 2: Training on unsegmented/segmented objects

Results from experiment 1 suggested that information from the background is present in the representation of the object, predominantly for more shallow networks. What happens if we train the networks on segmented objects, when all features are related to the object? To further explore the role of segmentation on learning, we trained ResNets differing in depth on a dataset with objects that were already segmented, and a dataset in which they were intact (i.e. embedded in a scene).

**Networks.**   As in experiment 1, we used deep residual network architectures (ResNets; [30]) with increasing number of layers (6, 10, 18, 34). Networks were implemented using the Keras and Theano code libraries [54,55]. In this implementation, input images were 128x128 randomly cropped from a resized image. We did not use ResNets with more than 34 layers, as the simplicity of the task leads to overfitting problems for the 'ultra-deep' networks.

**Stimuli.**   To train the networks, a subset of images from 10 different categories were selected from ImageNet. The categories were: bird 1 t/m 7, elephant, zebra, horse. Using multiple different types of birds helped us to increase task difficulty, enforcing the networks to learn specific features for each class (Table 1). The remaining (bigger) animals were added for diversity. From this subselection, we generated two image sets: one in which the objects were segmented, and one with the original images (objects embedded in scenes). Because many images are needed to train the models, objects were segmented using a DCNN pretrained on the MS COCO dataset [56], using the Mask R-CNN method [57] (instead of manually). Images with object probability scores lower than 0.98 were discarded, to minimize the risk of selecting images with low quality or containing the wrong object. All images were resized to 128x128 pixels. In total, the image set contained ~9000 images. 80% of these images was used for training, 20% was used for validation.

**Table 1. Dataset classes (categories) and the number of training and test stimuli.** Multiple different types of birds increased task difficulty, enforcing the models to learn specific features for each class. The remaining (bigger) animals were added for diversity.

| Category | train | test |
| --- | --- | --- |
| Bird 1 | 944 | 236 |
| Bird 2 | 867 | 217 |
| Bird 3 | 312 | 227 |
| Bird 4 | 455 | 114 |
| Bird 5 | 421 | 105 |
| Bird 6 | 930 | 233 |
| Bird 7 | 462 | 241 |
| Elephant | 700 | 175 |
| Horse | 290 | 72 |
| Zebra | 316 | 230 |

### Experimental procedure

First, we trained the four different ResNets for 100 epochs and monitored their accuracy after each epoch on the validation sets. Then, we reinitialized the networks with different seeds and repeated the process for 10 different seeds to obtain statistical results.

## Supporting information

**S1 Fig. Human performance (% correct) on the object recognition task, using centered 3D rendered objects on white, congruent of incongruent backgrounds.** Performance was higher for the segmented condition compared to congruent and incongruent.
(TIF)

## Acknowledgments

We thank Yannick Vinkesteijn for help with data collection for the human object recognition task, Sara Jahfari for helpful comments on the manuscript and all members of the Scholte lab for discussion.

## Author Contributions

**Conceptualization:** Noor Seijdel, Nikos Tsakmakidis, Edward H. F. de Haan, Sander M. Bohte, H. Steven Scholte.

**Data curation:** Noor Seijdel, Nikos Tsakmakidis.

**Formal analysis:** Noor Seijdel, Nikos Tsakmakidis.

**Funding acquisition:** Edward H. F. de Haan.

**Investigation:** Noor Seijdel, Nikos Tsakmakidis, Edward H. F. de Haan, H. Steven Scholte.

**Methodology:** Noor Seijdel, Nikos Tsakmakidis, Sander M. Bohte, H. Steven Scholte.

**Project administration:** Noor Seijdel, Sander M. Bohte, H. Steven Scholte.

**Resources:** Noor Seijdel, Edward H. F. de Haan, Sander M. Bohte, H. Steven Scholte.

**Software:** Noor Seijdel, Sander M. Bohte, H. Steven Scholte.

**Supervision:** Edward H. F. de Haan, Sander M. Bohte, H. Steven Scholte.

**Validation:** Noor Seijdel, Nikos Tsakmakidis.

**Visualization:** Noor Seijdel, Nikos Tsakmakidis.

**Writing – original draft:** Noor Seijdel, Nikos Tsakmakidis.

**Writing – review & editing:** Noor Seijdel, Nikos Tsakmakidis, Edward H. F. de Haan, Sander M. Bohte, H. Steven Scholte.

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
