## [Decision Letter · Decision Letter 0]

10 Feb 2020

Dear Ms Seijdel,

Thank you very much for submitting your manuscript "Depth in convolutional neural networks solves scene segmentation" for consideration at PLOS Computational Biology.

As with all papers reviewed by the journal, your manuscript was reviewed by members of the editorial board and by several independent reviewers. In light of the reviews (below this email), we would like to invite the resubmission of a significantly-revised version that takes into account the reviewers' comments.

We cannot make any decision about publication until we have seen the revised manuscript and your response to the reviewers' comments. Your revised manuscript is also likely to be sent to reviewers for further evaluation.

Sincerely,

Leyla Isik

Associate Editor

PLOS Computational Biology

Samuel Gershman

Deputy Editor

PLOS Computational Biology

Reviewer's Responses to Questions

**Comments to the Authors:**

Reviewer #1: Seijdel and colleagues conducted a study to test whether object information is differentiated from backgrounds and asked how it may work. The authors performed a behavioral experiment on humans and experiments on DNNs. The main finding is that DNN depth facilitates the segmentation of an object from a background. The main advancement in the study is showing how humans and DNNs differ and how deeper DNN may perform better on this task.

I see several issues with this manuscript that I would invite the authors to address.

1. The authors try to make claims about the recurrence being important for segmentation as it has been shown that deep DNNs can somewhat approximate recurrent DNNs. However, a more direct way to make this argument would be to test a recurrent DNN. One class of recurrent models (CORnet-R and CORnet-S) can be easily tested by extracting the activations using this Github repo: https://github.com/dicarlolab/CORnet. Additionally, other labs like Thomas Serre, Dan Yamins, Tim Kietzmann and Niko Kriegeskorte have recurrent models that could also be tested. If this manuscript is making claims about recurrent processing at least one recurrent model should be tested.

2. Differences between DNNs trained with segmented vs unsegmented images seem to be very small. As there are differences between DNNs trained multiple times I would like to ask the authors to train each DNNs tested at least three times with different initialization conditions to see whether the effects of visual training diet will be indeed significant across multiple initialization conditions of the models. All DNN figures should be based on multiple initialization conditions of the models, rather than just one, with error bars expressing standard deviation across initialization conditions.

3. Is seems that the numbers of images shown to participants and DNNs were different:

For humans: “243 images were generated for the actual experiment”

For DNNs: “810 images with a congruent background, 810 with an incongruent background and 270 images with segmented objects”

Why DNNs did not see the same images as humans to make the comparison between humans and DNNs more valid?

4. Figure 1D – What are the error bars? Std across participants? It should be stated in the figure legend.

5. Figure 2B – I would like to know which differences between the segmented, congruent and incongruent conditions are significant (like in Figure 1B for human participants).

6. Figure 5B - What do error bars represent? It should be stated in the figure legend.

7. In the discussion section there is a part about attention that I find quite confusing: “It also suggests that, with adequate deployment of attention, a deeper network is not necessary to recognize the object “. The authors should more clearly define what they mean by attention, how attention differs from recurrent processing, and how relevant it is to mention it here.

Reviewer #2: The authors have explored a timely topic re the depth of the DCNNs and the effect of depth on automatic scene segmentation. It was further interesting to see how this is potentially linked to the hierarchy of vision and the two modes of processing in the brain: feedforward vs. recurrent. I like their careful consideration of congruent and incongruent background, while some of the key previous literature has unfortunately ignored this important parameter by placing objects on incongruent backgrounds, when defining the core object recognition (see for example ’How does the brain solve visual object recognition?’ Neuron, 2012).

Major comments:

-Not sure what is the journal requirements, but starting off with the actual results without explaining the experiment itself was confusing. So please either move the methods before the results section; or otherwise integrate some of the method within the results (e.g. explain what experiment 1 is before jumping into the accuracy of participants in a task that is not explained before), and keep further details for the method.

-Figure 1, panel D: Please use non-parametric tests for comparing human accuracies (e.g. bootstrap of participants) —I am not convinced by the ANoVA. Do report the stats for all the pairwise comparisons. And explain what the error bars are ? Standard error? Std? Confidence interval ..? (ideally you would want to report 95% confidence intervals)

-Figure 2, panel B is one of the key results/figures, based on which most of arguments in the manuscript are formulated. However the results (and claims) here are missing a proper statistical support. E.g. it is said that ‘For shallow networks, performance is better for the congruent than for the incongruent condition’. What are the statistical analysis that support this argument? I suggest a non-parametric statistical test here to see if indeed the performance of shallower resnets is higher in congruent compared to incongruent —and report the p-value. Also consider multiple comparison correction (e.g. FDR). And similarly all other claims through out the paper that are related to the results of this figure need to be backed statistically.

-Figure 3, I could not find a good explanation of how interference (y-axis) is defined here. Please make sure this is explained in the figure legend and the method section.

-Page 12: “Models trained on segmented objects achieve better classification accuracy in the early stages” . There is no statical support for this statement (and the difference —by eyeballing— seems to be negligible)

- In the discussion, would be good to further explain and give insights that based on the results of this study, how many layers is deep enough for segmentation and give a high-level summary of what you promised in the abstract: “how, and when object information is differentiated from the backgrounds they appear on"

-It can very well enrich the paper if you provide visualisation of the deep net layers; and give an idea re the extracted features in each of the scenarios you lay out in experiment 2.

Minor:

-The link to the code and data (on the cover page) is broken. I could find the right page by google, but please update the hyperlink.

-Page 3, “Disruption of visual processing beyond feed-forward stages (e.g. >220 ms after stimulus onset, or after activation of higher order areas)”. : Most of the feedforward processing is done primarily within the first 150 ms after the stimulus onset. 220 ms is not accurate . Please see Liu et al. Neuron (2009), Cichy et al. , nature-neuro (2014), or Khaligh-Razavi et al. JoCN (2018)

-page 5: “This was confirmed by the observation that more shallow networks benefit more..” . two instances of ‘more’ ; remove the first one.

Page 16: ”For more complex scenes, on the other hand, the first feed-forward sweep might not be not sufficiently informative, …” . The second ‘not’ is unnecessary.

Page 19: “Participants performed on an object recognition task (Figure 1C).” ‘On’ is not needed.

**Have all data underlying the figures and results presented in the manuscript been provided?**

Reviewer #1: Yes

Reviewer #2: Yes

PLOS authors have the option to publish the peer review history of their article (what does this mean?). If published, this will include your full peer review and any attached files.

Reviewer #1: No

Reviewer #2: No
---

## [Decision Letter · Decision Letter 1]

6 Jun 2020

Dear Ms Seijdel,

We are pleased to inform you that your manuscript 'Depth in convolutional neural networks solves scene segmentation' has been provisionally accepted for publication in PLOS Computational Biology.

Before your manuscript can be formally accepted you will need to complete some formatting changes, which you will receive in a follow up email. A member of our team will be in touch with a set of requests. Also, please address the one lingering comment from the reviewer.

Best regards,

Samuel J. Gershman

Deputy Editor

PLOS Computational Biology

Reviewer's Responses to Questions

**Comments to the Authors:**

Reviewer #1: The authors have addressed all my comments.

Reviewer #2: The authors have properly revised the manuscript and my questions are addressed.

One minor issue for Figure 2: "Significant differences are indicated with a solid line [vs. a dashed line]" . This type of visualisation does not seem to cover the potential differences between 'incongruent' and 'segmented', as these two are not connected with any lines.

**Have all data underlying the figures and results presented in the manuscript been provided?**

Reviewer #1: None

Reviewer #2: Yes

PLOS authors have the option to publish the peer review history of their article (what does this mean?). If published, this will include your full peer review and any attached files.

Reviewer #1: No

Reviewer #2: Yes: Seyed Khaligh-Razavi

---

## [Editor Report · Acceptance letter]

16 Jul 2020

PCOMPBIOL-D-19-02064R1 

Depth in convolutional neural networks solves scene segmentation

Dear Dr Seijdel,

I am pleased to inform you that your manuscript has been formally accepted for publication in PLOS Computational Biology. Your manuscript is now with our production department and you will be notified of the publication date in due course.

With kind regards,

Laura Mallard
